

# Evaluating alternative hypotheses to explain the downward trend in the indices of the COVID-19 pandemic death rate

Sonali Shinde[1], Pratima Ranade[1] and Milind Watve[2]

[1] Department of Biodiversity, Abasaheb Garware College, Pune, Pune, Maharashtra, India
[2] Independent Researcher, Pune, Maharashtra, India

## ABSTRACT

**Background:** In the ongoing Covid-19 pandemic, in the global data on the case fatality ratio (CFR) and other indices reflecting death rate, there is a consistent downward trend from mid-April to mid-November. The downward trend can be an illusion caused by biases and limitations of data or it could faithfully reflect a declining death rate. A variety of explanations for this trend are possible, but a systematic analysis of the testable predictions of the alternative hypotheses has not yet been attempted.

**Methodology:** We state six testable alternative hypotheses, analyze their testable predictions using public domain data and evaluate their relative contributions to the downward trend.

**Results:** We show that a decline in the death rate is real; changing age structure of the infected population and evolution of the virus towards reduced virulence are the most supported hypotheses and together contribute to major part of the trend. The testable predictions from other explanations including altered testing efficiency, time lag, improved treatment protocols and herd immunity are not consistently supported, or do not appear to make a major contribution to this trend although they may influence some other patterns of the epidemic.

**Conclusion:** The fatality of the infection showed a robust declining time trend between mid April to mid November. Changing age class of the infected and decreasing virulence of the pathogen were found to be the strongest contributors to the trend.

Corresponding author
Milind Watve,
milind.watve@gmail.com

## INTRODUCTION

A consistent global trend in the current Covid-19 pandemic is that of decreasing case fatality rate (CFR). Whether this is an illusion created by some biases and limitations of data and if not, what are the possible alternative causes of the decline is the question we address here. In an on-going epidemic, an estimate of true death rate is rather difficult for several reasons including inappropriate diagnosis, time lag in diagnosis and mortality, and rapidly changing population dynamics. Nevertheless, a number of indices can reflect death rates with some limitations. The case fatality rate, a cumulative index of the

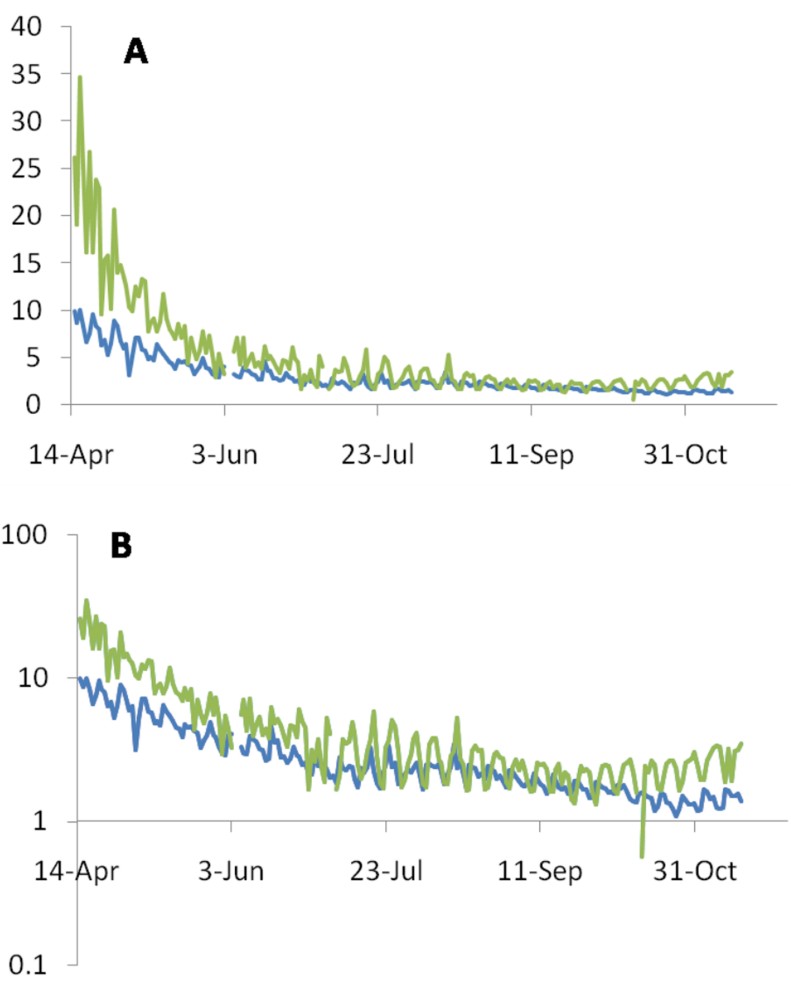

**Figure 1 Consistent monotonic decline by a factor of 4 to 5 in the ND/NC (blue line) and ND/NR (green line) ratios expressed as percentages between mid-April to mid-November.** (A) On the linear scale the decline appears to be saturated after July, but the subtle continued decline is revealed by the log scale (B).                                     

number of deaths attributed to the virus divided by the confirmed positive cases so far, is most commonly used (*Ghani et al., 2005*; *Reich et al., 2012*). This index has shown a decline in global data [1,2]. (Note that we use square brackets for citing public domain data). However, being cumulative, this index is less sensitive to time trends and is dominated by the phase having more number of cases. We therefore use two other ratios here which are more faithful to the time trend, although somewhat more sensitive to stochasticity. The ratio of the number of new deaths in a day (ND) to the number of new cases registered (NC) on that day (ND/NC) is one index and the ratio of ND to the number declared recovered (NR) on that day (ND/NR) the other. The pros and cons of these indices are detailed in the Supplemental Material.

It can be seen in the global picture that from mid-April to mid-November, both the ratios show a consistent monotonic decline although not quite linearly [1, 2, 3] (Fig. 1). The ND/NC ratio was close to 10 in mid-April, which came down to between 1 and 2

by mid-November. ND/NR also declined in similar proportion. So among the recorded global data there is a 4 to 5 fold difference in these indices between mid-April and mid-November.

It is well recognized from the early phase of the epidemic that CFR is a substantial underestimate of the infection fatality rate (IFR). IFR estimates the proportion of death from infected individuals [7] not all of which may be detected and diagnosed. There have been a number of attempts to estimate the IFR which are consistently smaller than CFR. However, the methods of estimating IFR vary, the IFR estimates are spatially and temporally fragmentary and therefore plotting time trends in IFR is not possible. If we assume that the bias in CFR is more or less constant over time, the time trend in CFR may reflect the time trend in IFR as well. However, it is quite possible that the bias itself has an increasing or decreasing time trend. Therefore it is necessary to test for a temporal trend in the bias as well. The IFR estimates till June were between the ranges of 0.09% to 1.6% with a mean of 0.68% (*Meyerowitz-Katz & Merone, 2020*). However, in later serosurveys the estimate of the incidence increases substantially (*Bobrovitz et al., 2020*; *Bhattacharyya et al., 2020*). Therefore it is possible that even IFR is much smaller than the one estimated earlier or is declining considerably in time similar to CFR.

While underreporting of deaths has also been a serious problem, death underreporting has been disproportionately smaller than case underreporting (*Bhattacharyya et al., 2020*; *Chakravarty, 2020*). Therefore any correction for the reporting bias will reveal a death rate much lower than the CFR. Using CFR along with the ND/NC and ND/NR ratios, wherever appropriate, we will first list the possible explanations, examine the differential testable predictions and evaluate their relative contributions in the declining trend. Further, since the different explanations are not mutually exclusive, we explore the possibility of their interactions.

## The alternative hypotheses

**A.** We first consider the possibility that the downward trend is illusionary for one or more of the following reasons.

(i) **Time lag in diagnosis and death:** Because of the inevitable but unpredictable time lag, by the time deaths are recorded, the number of cases might have gone up and therefore the indices of death rates are an underrepresentation of true death rate.

(ii) **Increased testing detected more asymptomatic cases:** The assumption behind this explanation is that the death rate was always as low as it is apparent today. The IFR that can be calculated from recent serosurveys is of the order of 0.02% to 0.07% (*Bhattacharyya et al., 2020*; *Ioannidis, 2020*; *Pune Municipal Corporation, 2020*). If this was the true death rate right from the beginning, then the social implication is serious. The perceived death rate was the basis on which a number of measures were imposed by different state administrations, which have seriously affected the livelihood of a large population. If people develop an impression that it was a false alarm, they may lose trust in international and national health authorities including WHO. This may have serious

long term consequences. Therefore it is extremely important to evaluate this possibility carefully.

(iii) **The age class of patients changed:** Covid 19 is known to cause disproportionately higher deaths in the elderly (*Ioannidis, 2020*; *Bonanad et al., 2020*; *Williamson et al., 2020*). So if more individuals of the younger age class are infected in the later phases of the epidemic, there would be an apparent decline in the indices of death rates.

**B.** An alternative possibility is that the downward trend in death rate is real and because of one or more of the following reasons.

(i) Increased efficiency of treatment regime brings down the death rate.

(ii) Increased immunity in the population reduces the death rate.

(iii) The virus progressively lost its virulence: A number of evolutionary epidemiology models indicate the possibility that under certain set of conditions progressive evolution of a newly invading virus leads to reduced virulence (*Ewald, 1994*; *Anderson & May, 1982*; *Read, 1994*; *Levin & Bull, 1994*; *Kerr et al., 2012*). This might be applicable to Covid-19.

We would now comparatively evaluate the alternative hypotheses for the apparent decline in the indices of death rate using data from public domain.

## METHODOLOGY

### Sources of data

We use data from sources available in the public domain, mainly from WHO, CDC [1, 7, 8] and other open sources giving raw data as well as patterns seen in it. A systematic review was unwarranted because data were available from very standard and trusted sources such as WHO, CDC and other official national and international sources. The other sources mainly include World meter [3] Our World in Data [2], and Covid19India [4].

The data sources are listed below and specifically cited in the text with the appropriate hyperlink. There are certain inevitable limitations in the data. Data collection from different countries has subtle differences in the method of collection and accordingly some inevitable biases. We use pooled global data whenever available but some numbers are not available from all countries. For example, data on the number of tests performed is not available globally. Furthermore some countries report the number of tests performed and others report the number of individuals tested. The two are not inter-convertible and their implications can be different. Whenever, getting global figures is not possible, we take the four countries with maximum number of cases reported so far, namely the United States, Brazil, India and Russia and perform country specific analysis.

### Testable predictions and evaluation of the hypotheses

**Hypothesis A(i): The time-lag between diagnosis, death and recovery:** This leads to under or over-estimates of death rates. Simultaneous use of two ratios, ND/NC and ND/NR (Fig. 1) can resolve the issue (see Supplemental Material S1). Since in the global data we see a consistent decline in both the ratios, it is unlikely to be an illusion created by
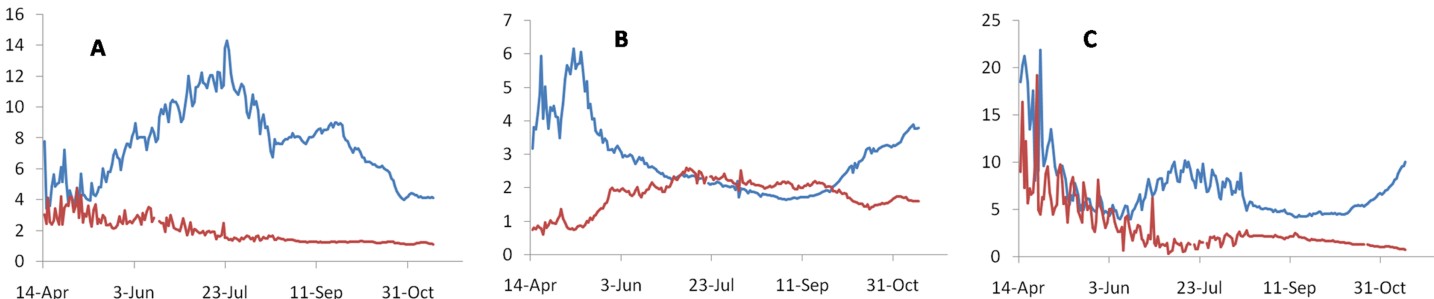

**Figure 2** Time trends in the proportion of positives detected per day during the testing effort (blue lines), which inversely reflects the testing efforts and the ND/NC ratio (red lines). Since global data on testing are not available, we use three countries with maximum number of cases reported. This includes India (A), Russia (B) and United States (C). If increased testing efficiency was mainly responsible for the apparent downward trend in death rate, the time trends in both the curves should have matched at least qualitatively, i.e., a downward trend in the proportion of positives should be matched by a downward trend in ND/NC. Observed trends do not support this prediction consistently.

the time lag effect. Further along the course of the epidemic, Rt, which measures the virus transmission rate has been declining globally (*Contreras et al., 2020*). When Rt is declining, ND/NC tends to be overestimated. A decline in ND/NC in spite of a declining Rt is a robust indication of a true decline. Therefore the time lag bias alone is unable to explain the consistent decline in ND/NC.

**Hypothesis A(ii): Increased testing detected more asymptomatic cases:** If we assume that the death rate was always low but in the initial phases of the epidemic, it was overestimated because of inadequate testing efficiency. If the testing effort effectively increased, the proportion of positives among the tested could have decreased. In contact tracing, the number tested is expected to increase with the number of positive cases found at a given time. Therefore rather than the absolute number of tests or the number of individuals tested, the proportion of positives is a better inverse indicator of the effective testing effort.

Using this principle, a testable prediction of this hypothesis is that there should be a positive correlation between the proportion of positives among the tested and ND/NC. If changing testing effort is the main reason for apparent change in death rates, then the time trend in the proportion of positives should match the time trend in the ratio at least qualitatively. That is if the proportion of positives is declining, the ratio should also decline. Unfortunately global data on the number of tests are not available. In country level analysis of the focal countries reporting maximum incidence, namely India, US and Russia we find different and mutually inconsistent patterns (Fig. 2). Across 91countries for which the testing data at least as recent as August 2020 was available at the time of analysis [5], We do not find a positive correlation between the proportion of positives and CFR which was expected if testing efficiency was the main factor deciding the apparent CFR (Fig. 3).

Furthermore, there have been many attempts to estimate the true proportion of infected individuals in a population vis-a-vis the registered cases. This is attempted using a variety of methods, at different locations and different phases of the epidemic. So a rigorous comparative analysis is not possible. Grossly, while the estimated seroprevalences during

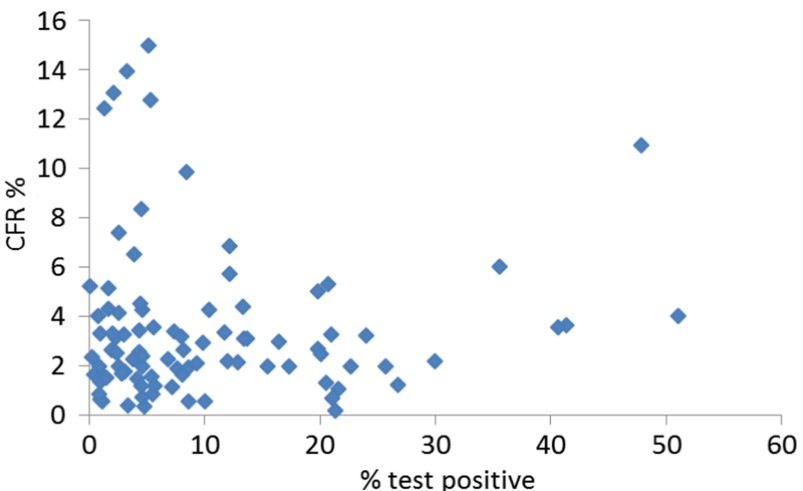

**Figure 3 Correlation between positive tests and CFR.** If the true death rate was more or less constant spatiotemporally but the apparent trends were caused by testing biases, we would expect a positive correlation between the proportion tested positive and the CFR. In data on 91 countries for which test data were available and which had at least 100 deaths, the expected correlation is not seen ($r^2 = 0.0004$, NS).

the trials completed by April are all less than 10% with only one exception from Iran (*Bobrovitz et al., 2020*), the July and August trials have estimates in the range of 15% to 57% (*Pune Municipal Corporation, 2020*; *Chakravarty, 2020*; *Majiya et al., 2020*). In areas where a comparison of the fold increase in cases to fold increase in seroprevalence during the same period is possible, in some countries such as in US [8] (*Havers et al., 2020*; *Sutton, Cieslak & Linder, 2020*) and Brazil (*Silveira et al., 2020*) the ratio of the rise in seroprevalence to rise in cases declined with time as expected by the hypothesis. But in many other countries it increased (*Pune Municipal Corporation, 2020*; *Chakravarty, 2020*; *Murhekar et al., 2020*; *Nisar et al., 2020*; *Stringhini et al., 2020*; *BBC News, 2020*). Collectively there is no consistent evidence that the overestimation bias in the CFR and ND/NC ratio reduced with time. Thus there is no consistent evidence that the bias in CFR estimates has been consistently reducing with time. Therefore the hypothesis that the downward trend is caused by a greater bias in the earlier phases of the epidemic and gradual removal of the bias subsequently is not supported by evidence.

**Hypothesis A (iii): The age class of patient changed:** The age class distribution among the diagnosed cases has evidently changed with time in global data [6]. Also the differential case fatality across age classes is well known (*Ioannidis, 2020*; *Bonanad et al., 2020*; *Williamson et al., 2020*). Therefore it is very likely that at least qualitatively the changed age class distribution may explain the apparent decline in death rates. We need to estimate to what extent the changed age distribution explains the decline.

The >65 age group has declined from 28% to 10% among the infected population between mid-April to end-July during which time ND/NC in the registered cases declined by 75–80% to remain at 20–25%. If we take a limiting assumption that all deaths are only in the >65 group, the death rate would have declined by 64% to come down to 36%. By this assumption a changing age distribution explains a substantial part but not the

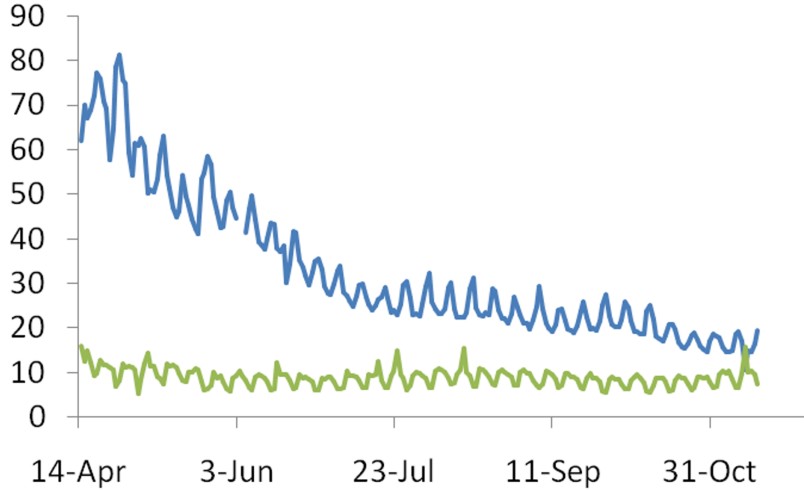

**Figure 4 The time trend in the ratio of patients under critical care on a given day to the number of cases recorded on that day (blue line) and percentage deaths under critical care (green line).** The decline in death rates appears to be more due to decline in the proportion of serious cases needing critical care than due to success rate in critical care.

entire reduction in death rate. This is a limiting estimate assuming all deaths are in the >65 age class. The age class distribution of deaths is different in different countries since the age class distribution of the population itself is widely different. In the US, about 20% deaths were in the class < 65 and in India, 47% deaths were among the <60% age class [9]. Considering that the 64% reduction was only in the >65 or >60 class, the expected decline is 51.2% and 33.92% in the two countries respectively. This indicates that changing age class distribution may explain a substantial part of the apparent decline in death rate but still leaves a considerable decline unexplained. Further among the elderly age class itself, up to 2/3$^{rd}$decline in death rate is shown in some countries [10] which also means that the decline is not entirely contributed by changing age distribution.

**Hypothesis B(i): Increased efficiency of treatment regime brings down the death rate:** Global trends show that there is substantial reduction in the proportion of patients under critical care but there is only a marginal or no reduction in the proportion of deaths among patients under critical care (Fig. 4) [2].

This need not necessarily mean that there is no improvement in treatment success. It is likely that the age distribution of the hospitalized has itself changed. If the mean age in this class has gone up, this may mask the expected improvement in the treatment. Even in that case the success rate of treatment does not explain the downward trend in death rates. Moreover the documented improvements in the success rates are limited. Clinical trials of all mainstream drugs have failed to show any significant reduction in mortality (*WHO Solidarity Trial Consortium, 2020*; *Spinner et al., 2020*; *Beigel et al., 2020*; *Li et al., 2020*; *Burki, 2020*; *Mitjà et al., 2020*; *Simonovich et al., 2020*). More relevant for our question is the fact that there is a considerable decline in the proportion needing critical care. Rather than improvement in treatment results, the need for treatment itself appears to have gone down.

**Hypothesis B(ii): Increased immunity in the population reduces the death rate:**
Going by the registered cases a very small fraction of the population is exposed to the
infection to see any major population level immunity change. However, going by
seroprevalence, a much larger fraction of the population appears to have been sub clinically
infected. It is possible that a substantial fraction of the population has indeed been exposed
and presumably became immune. This immunity may be partially responsible for the
reduced mortality. However, a critical question here is what made the large proportion of
asymptomatic cases possible? If the virus was as virulent as initially perceived, the epidemic
wouldn't have progressed to cause so many asymptomatic infections. The low mortality
and mild clinical course therefore should be a cause, rather than a consequence of the
increasing population immunity (*Johansson et al., 2021*).

**Hypothesis B(iii): The virus progressively lost its virulence:** The concept that in the
process of host-parasite co-evolution, a pathogen often evolves towards reduced virulence
is quite old, but evolution towards loss of virulence is conditional. Not all pathogens
appear to have reduced virulence when they coexist with a host population for a long time
(*Kerr et al., 2012*; *Ewald, 1994*; *Anderson & May, 1982*; *Bull, 1994*). Many evolutionary
epidemiology models for optimum virulence were built and the continued theoretical
development was backed up by epidemiological (*Cressler et al., 2016*; *Best & Kerr, 2000*;
*Kale, Chanda & Watve, 2002*) as well as experimental studies (*Berngruber et al., 2013*;
*Tardy et al., 2019*). With respect to the Covid-19 pandemic there are a multitude of reasons
why evolution towards reduced virulence can be expected.

A fundamental assumption behind the evolution of virulence models is that the severity
of symptoms and fatality is at least partly decided by the virulence of the virus. On the
other hand, a virus that kills its host rapidly, gets less time to spread from the infected
individual. Secondly the quarantine measures applied all over the world are likely to have
created a selective force upon the virus. Since a symptomatic case is more likely to
undergo testing and subsequently quarantined, a virulent variant causing more serious
symptoms is more likely to be quarantined. A milder variant, has a greater chance of
escaping detection and subsequent quarantine. Thirdly, if virulence is tightly correlated to
viral loads and thereby transmission success, the virulent variant can have a greater
selective advantage (*Ewald, 1994*; *Read, 1994*; *Anderson & May, 1982*). If virulence does
not have a direct correlation with infection intensity and pathogen transmission, it is
likely to be selected against (*Levin & Bull, 1994*; *Read, 1994*). If the viral loads are not
consistently higher in serious cases, this advantage can be assumed to be marginal and not
sufficient to compensate the quarantine disadvantage. In SARS Cov-2 infections there is a
large overlap in the viral loads of symptomatic or fatal versus asymptomatic cases and
even in studies where there is a statistically significant difference, the effect size or the
magnitude of difference is not very large (*Lennon et al., 2020*; *Pujadas et al., 2020*). Some
mutants such as Gly614 showed higher viral loads but are not more virulent (*Long et al.,
2020*). The absence of viral load and virulence correlation makes this virus an ideal
case for evolution towards reduced virulence. Further at least some components of the
immune response are expressed in proportion to the extent of invasion by the pathogen
(*Read, 1994*; *Spooner & Yilmaz, 2011*). If the host response is proportional to the extent of

invasion, a milder virus may survive better in a more resistant host, while a virulent one may do better in a susceptible host. If this is true, host immunity and viral virulence are expected to interact in a positive feedback loop. As the population acquires greater immunity, a milder virus can experience a selective advantage. Thus there are multiple reasons why SARS-Cov-2 may have experienced a selective pressure for reduced virulence.

Rejection or quantitatively inadequate explanation by other hypotheses is an indirect support to the evolution hypothesis. But a true test of the hypothesis is to show evolutionary changes in the genome indicating reduced virulence. Over 50,000 genomes have been sequenced in various parts of the world during the pandemic. There are on an average over 7 mutations per genome (*Van Dorp et al., 2020*; *Mercatelli & Giorgi, 2020*). So the mutation rate can be assumed to be sufficient to generate the required variation for natural selection. Among the single nucleotide mutations there is a high proportion of recurrent non-synonymous mutations (*Spooner & Yilmaz, 2011*). The distribution of mutations along the genome is highly non-random and the genes responsible for the pathogenicity have accumulated significantly higher frequency of mutations (*Choi et al., 2020*). Both these patterns indicate strong selective pressure for infectivity and/or virulence related genes. Genomic signatures of strong selection coupled with the declining death rates not explained completely by other hypotheses makes the evolution towards reduced virulence hypothesis more promising. One of the mutations, D614G is suspected to increase the cell adhesion but whether it affects the infectivity or virulence or both is not clearly known (*Mercatelli & Giorgi, 2020*; *Brufsky, 2020*). On the contrary, there are many other mutations in the region of the spike protein S1, S2 and docking studies show that they reduce the stability of the host cell binding complex. Furthermore in a comparative study of four regions of India, the ones with lower average stability of mutants in the spike protein correlated negatively with local CFR (*Banerjee et al., 2020*). These mutations are likely candidates responsible for the loss of virulence. There are many other mutations in structural and non-structural proteins (*Paul et al., 2020*) which are also likely to play a role in determining virulence. Virulence is a complex phenomenon and from previous studies it is apparent that a large number of genes may contribute to viral virulence (*Kerr et al., 2012*). Unfortunately as yet we do not have sufficient knowledge linking specific mutations to their phenotypic effects. There is no standardized empirical test of virulence to examine the effects of specific mutations on virulence. Of more direct relevance is the observation that the mutational set observed among samples coming from symptomatic versus asymptomatic cases is significantly different (*Paul et al., 2020*). This is the most direct indicator that the asymptomatic clinical course is likely to be at least partly driven by changes in the viral genome.

## CONCLUSION

In summary, based on the available evidence, the hypotheses that there is a time lag in diagnosis and death [A(i)]and increased testing detected more symptomatic cases [A(ii)] fail to get any supportive evidence, increased efficiency of treatment regime brings down the death rate [B(i)] does not appear to have made a strong contribution to the trend. The change in age class of patients [A(iii)], death rate reduction due to increased immunity

in the population [B(ii)] and loss of virulence [B(iii)] are likely to be important causes of the decline in death rates, out of which [B(ii)] increased immunity in the population needs prior initiation by either or both the others. Therefore a combination of changed age class [A(iii)] and loss of virulence [B(iii)] are most likely the primary causes of the declining trend and a combination of both appears to be necessary to explain the trend quantitatively. The three are not mutually exclusive and in fact may interact with each other. The cause of changed age class in the infected population could be that the older age classes are being effectively protected by the prevalent preventive measures, but it is also likely that the virus has evolved to infect younger age classes. It is of particular relevance here that some of the mutants are non-randomly represented in different age classes (*Paul et al., 2020*). This is possible if certain mutants are more likely to invade younger age classes.

An infectious epidemic is a complex process and multiple factors decide the outcomes. Our efforts to differentially support alternative hypothesis is limited by some constraints on data. We did not find a single data set giving all necessary parameters. Data on some variables is not available from all countries. Nevertheless, despite these limitations, the emerging patterns are strong enough to support some of the hypotheses. Previous studies on the changing parameters of an epidemic have mostly been published post facto (e.g., *Best & Kerr, 2000*; *Kale, Chanda & Watve, 2002*; *Kerr et al., 2012*). Analyzing patterns during an ongoing epidemic has a different set of challenges. Therefore we used a set of indices and methods different from earlier studies. But the trend in apparent reduction in virulence is similar to the earlier findings (*Best & Kerr, 2000*; *Kale, Chanda & Watve, 2002*; *Kerr et al., 2012*).

The specific contribution of our analysis is to suggest the evolutionary changes in a virus can be detected during an ongoing epidemic and policies to minimize population loss should take into account the ongoing evolution. Understanding the nature of selection on the virus can lead to virulence management strategies. Viruses have short generation times and high mutation rates and therefore can evolve very fast. Evolution on the background of host physiology, immunity, behavior, public health policies and available treatments should be an intrinsic part of epidemiological theories and models, which is likely to deepen our understanding of the disease process at different levels.

### Funding
The authors received no funding for this work.

### Competing Interests
The authors declare that they have no competing interests.

### Author Contributions
- Sonali Shinde performed the experiments, analyzed the data, prepared figures and/or tables, authored or reviewed drafts of the paper, and approved the final draft.

- Pratima Ranade performed the experiments, analyzed the data, prepared figures and/or tables, and approved the final draft.
- Milind Watve conceived and designed the experiments, performed the experiments, analyzed the data, prepared figures and/or tables, authored or reviewed drafts of the paper, and approved the final draft.

## Data Availability

We used data that are already in the public domain.

[1] WHO Coronavirus disease (COVID-19) Weekly Epidemiological Update and Weekly Operational Update https://www.who.int/emergencies/diseases/novel-coronavirus-2019/situation-reports.

[2] Our world in data: Statistics and research. Mortality rate in Covid-19. https://ourworldindata.org/mortality-risk-covid.

[3] Worldometer: Coronavirus worldwidegraphs https://www.worldometers.info/coronavirus/worldwide-graphs/.

[4] WHO Corona Disease (COVID 19).Covid19India.org.

[5] Wikipedia: Covid-19 testing https://en.wikipedia.org/wiki/COVID-19_testing.

[6] WHO coronavirus situation report, 5th August 2020 https://www.who.int/docs/default-source/coronaviruse/situation-reports/20200805-covid-19-sitrep-198.pdf?sfvrsn=f99d1754_2.

[7] World Health Organization. Estimating mortality from COVID-19. Available from https://www.who.int/news-room/commentaries/detail/estimating-mortality-from-covid-19.

[8] Interactive Serology Dashboard for Commercial LaboratorySurveys

https://www.cdc.gov/coronavirus/2019-ncov/cases-updates/commercial-labs-interactive-serology-dashboard.html.

[9] Age class distribution link https://www.hindustantimes.com/india-news/85-deaths-in-45-plus-age-bracket-govt-data/story-II4EFnB7APRnLuU3MPmyZK.html.

[10] DecliningCOVID-19CaseFatality Rates: https://www.cebm.net/covid-19/declining-covid-19-case-fatality-rates-across-all-ages-analysis-of-german-data.

## Supplemental Information

Supplemental information for this article can be found online at http://dx.doi.org/10.7717/peerj.11150#supplemental-information.

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
