# Peer review of "Evaluating alternative hypotheses to explain the downward trend in the indices of the COVID-19 pandemic death rate"

_PeerJ, doi:10.7717/peerj.11150_

## Round 0.1 · original submission · Major Revisions

After assessing your manuscript and the reviewers' comments, I think your work has scientific merit to be published in PeerJ, once some issues are solved by you. Please, see their reports below this letter.

Reviewer 1 ·

Basic reporting

The paper is, overall, well-written. However, the authors have assumed that the reader already has a relatively high background knowledge. It would be helpful if the introduction also explained some of the basics about recording of deaths from COVID-19 and the issues associated with this. In lines 36/37 it is stated that "an estimate of true death rate is rather difficult for several reasons." However, those reasons are not outlined (which they should be).

All terms should be defined, e.g. CFR, IFR, Rt.

The headings of the different hypotheses should be carried over to the results section as well. At the moment, the reader needs to refer back to earlier sections in order to remember which hypothesis is being discussed.

Experimental design

The experimental design is unusual in that a list of hypotheses is being tested using publicly-available information.

While the data sources have been identified in the main text, the actual original methodology is outlined in the supplementary files. I am unsure whether this is consistent with the style of the journal, but there at least needs to be some reference to the supplementary files.

It is unclear whether there was a systematic search for data, although the lack of a search strategy would indicate that this has not happened. This would be the expected standard to address the study questions. The authors need to clearly state why they haven't conducted a systematic review.

Validity of the findings

It is likely that the downward trend in COVID-19 death rate is multi-factorial, potentially contributed to by all the stated hypotheses. The authors state this and indicate the potential strength of support for each hypothesis. However, they also need to discuss the limitations of the study given the reliance on incomplete data. It would also be useful to discuss similar issues with previous newly-described infections.

Additional comments

The study question is important. This is a potentially useful paper in the context of the ongoing pandemic.

Reviewer 2 ·

Basic reporting

This is an interesting paper, but it will take a lot of modification for this paper

Experimental design

This is an interesting paper, but it will take a lot of modification for this paper

Validity of the findings

This is an interesting paper, but it will take a lot of modification for this paper

Additional comments

This is an interesting paper, but it will take a lot of modification for this paper. The questions are attached in the pdf file

Annotated reviews are not available for download in order to protect the identity of reviewers who chose to remain anonymous.

---

## Round 0.2 · Minor Revisions

Regarding the comments of reviewer #1, there are some points which are not correctly addressed, which you should assess in a new revised version of the text:

1. Where is the reference to the supplemental material in the text?

2. "It is unclear whether there was a systematic search for data, although the lack of a search strategy would indicate that this has not happened. This would be the expected standard to address the study questions. The authors need to clearly state why they haven't conducted a systematic review."

Although I agree with the response, no change have been done into the text.

3. "It is likely that the downward trend in COVID-19 death rate is multi-factorial, potentially contributed to by all the stated hypotheses. The authors state this and indicate the potential strength of support for each hypothesis. However, they also need to discuss the limitations of the study given the reliance on incomplete data. It would also be useful to discuss similar issues with previous newly-described infections."

Not correctly addressed.

Reviewer 2 ·

Basic reporting

See general comments.

Experimental design

See general comments.

Validity of the findings

See general comments.

Additional comments

The article has presented clearly and in detail to the readers. My questions have been satisfactorily resolved, so I suggest accepting this version.

---

## Round 0.3 · accepted · Accept

All the reviewers' concerns have been correctly addressed in this new revised version of the text.